# Tailoring Mechanical Properties of a-C:H:Cr Coatings

Alireza Bagherpour [1], Paul Baral [2,3], Marie-Stéphane Colla [2], Andrey Orekhov [2,4], Hosni Idrissi [2,4], Emile Haye [1,*], Thomas Pardoen [2,5] and Stéphane Lucas [1]

[1] Laboratoire d'Analyse par Réaction Nucléaires (LARN), Namur Institute of Structured Matter (NISM), University of Namur, 61 Rue de Bruxelles, 5000 Namur, Belgium; alireza.bagherpour@unamur.be (A.B.); stephane.lucas@unamur.be (S.L.)

[2] Institute of Mechanics, Materials and Civil Engineering (iMMC), Université catholique de Louvain, av G Lemaître 4, 1348 Louvain-la-Neuve, Belgium; marie-stephane.colla@uclouvain.be (M.-S.C.); andrey.orekhov@uantwerpen.be (A.O.); hosni.idrissi@uclouvain.be (H.I.); thomas.pardoen@uclouvain.be (T.P.)

[3] Univ. Lyon, Mines Saint Etienne, CNRS UMR 5307 LGF, Centre SMS, 42100 Saint Etienne, France

[4] Electron Microscopy for Materials Science (EMAT), University of Antwerp, Groenenborgerlaan 171, 2020 Antwerp, Belgium

[5] WEL Research Institute, Avenue Pasteur, 6, 1300 Wavre, Belgium

* Correspondence: emile.haye@unamur.be

**Abstract:** The development of coatings with tunable performances is critical to meet a wide range of technological applications each one with different requirements. Using the plasma-enhanced chemical vapor deposition (PECVD) process, scientists can create hydrogenated amorphous carbon coatings doped with metal (a-C:H:Me) with a broad range of mechanical properties, varying from those resembling polymers to ones resembling diamond. These diverse properties, without clear relations between the different families, make the material selection and optimization difficult but also very rich. An innovative approach is proposed here based on projected performance indices related to fracture energy, strength, and stiffness in order to classify and optimize a-C:H:Me coatings. Four different a-C:H:Cr coatings deposited by PECVD with $Ar/C_2H_2$ discharge under different bias voltage and pressures are investigated. A path is found to produce coatings with a selective critical energy release rate between 5–125 $J/m^2$ without compromising yield strength (1.6–2.7 GPa) and elastic limit ($\approx$0.05). Finally, fine-tuned coatings are categorized to meet desired applications under different testing conditions.

**Keywords:** chromium-doped hydrogenated amorphous carbon; magnetron sputtering; tensile testing; fracture toughness; hardness; materials selection

## 1. Introduction

Amorphous carbon (a-C) coatings have been extensively studied for their outstanding mechanical [1,2], chemical [3], and biological properties [4]. These coatings, deposited by vacuum technology, have shown tunable properties such as high hardness, low friction, and acceptable wear resistance. However, variations in coating properties, poor adhesion, thickness uniformity, and environmental stability remain challenging [5].

Plasma-Enhanced Chemical Vapor Deposition (PECVD) allows for tuning coatings' physical and chemical properties by varying deposition parameters, such as pressure and gas composition. In addition, the chemical structure is affected by the magnitude of the impinging energy of the ions [6]. Acetylene ($C_2H_2$) is often used for a-C:H deposition due to its high electron-molecule cross-section and high deposition rate [7,8].

Adding metal elements to the coatings can decrease internal stress and improve wear resistance, adhesion, thermal stability, and fracture toughness [9,10]. The rise in $sp^2$ graphitization by doping metal elements and partly hard carbide phase dispersion in amorphous carbon explains this phenomenon [11]. Among many metallic elements,

chromium (Cr), in combination with carbon, presents attractive mechanical properties (exceptionally stable friction performance and toughness) in a-C coatings [12–14].

The aim of this research is to understand further how the deposition parameters affect the mechanical characteristics (both effective and intrinsic) of thin a-C:H:Cr coatings with thickness in the 1 μm range to build a rational coating selection strategy to optimize the systems further. To this end, we capitalized on the previous work by our group [15,16] to demonstrate how a-C:H:Cr coatings can be produced with tailored mechanical properties, specifically strength and fracture toughness by varying only the deposition conditions in $C_2H_2$ magnetron-assisted discharge. Because of the ionic nature of the condensing species in such a setup, one can easily understand that substrate biasing is one of the methods to produce coatings with different properties, as well as traditional pressure variation. Four coatings have been selected as representative candidates for this study and characterized by comprehensive techniques, including X-ray Photoelectron Spectroscopy (XPS) to extract the coating's chemical composition; Atomic Force Microscopy (AFM) and Scanning Electron Microscopy (SEM) to observe the structural and morphological properties of the coatings; Transmission Electron Microscopy (TEM) and X-ray diffractometer (XRD) to investigate the amorphous microstructure of the coating; Raman Spectroscopy to track structural variation; nanoindentation to determine the hardness, Young's modulus, and activation volume; micro-scratch test to extract wear resistance, critical load, as well as tensile tests on a supporting polymer membrane to determine the fracture toughness of the coating. Comparison and deeper discussion are made through analyzing materials property map presenting the ratio of yield strength over Young's modulus ($\sigma_y/E'$) against critical fracture energy release rate ($G_{Ic}$) for several materials. The performances are shown to position extremely well compared to other available coatings for various applications asking for high-yielding strength, high critical load under scratching conditions, and elastic limit, with a wide range of critical energy release rates.

## 2. Materials and Methods

### 2.1. Coating Deposition

The a-C:H:Cr coatings with 10 at% of Cr and around 30 at% of hydrogen content were deposited on {100} silicon wafers, on 13 μm and 25 μm thick HN type Kapton substrates by means of PECVD magnetron sputtering [15,16]. Silicon substrates were ultrasonically cleaned in soap and de-ionized water and dried with hot air prior to the deposition. The polymer substrates were used as received by the supplier. A base pressure of $10^{-4}$ Pa was created prior to the deposition to remove contamination. Then, a 10 min Cr target cleaning with 1.2 kW power was performed under a pure Ar atmosphere. Cr targets (7.5 × 35 cm, purity 99.8%, Nova Fabrica) have been sputtered in argon and acetylene ($C_2H_2$) mixed atmosphere (150 sccm Ar/160 sccm $C_2H_2$) under different deposition conditions (0.66 Pa, 2.66 Pa sputtering pressure, floating or −100 V bias voltage). The gas flow rate was controlled by FloTron multi-channel reactive gas monitoring system. A bipolar pulsed power supply working with a frequency of 1250 Hz and a pulse duration of 300 μs on and 100 μs off was used (duty cycle of 75%). Three-fold sample rotation around the Z axis along with 2 rpm table rotation speed has been applied in order to generate homogeneous coating. The deposition was conducted at room temperature. The coating thickness was set at around 1 μm, i.e., sufficiently thick to perform the relevant mechanical tests. Figure 1 schematically illustrates a cross-section of the semi-industrial chamber of about one cubic meter from D&M Vacuumsystemen.

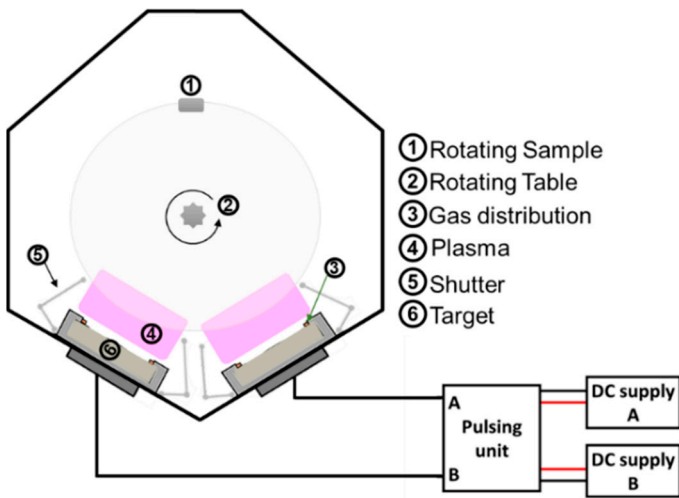

**Figure 1.** 1 m$^3$ semi-industrial deposition chamber.

*2.2. Characterization of the Coatings*

2.2.1. Chemical and Structural Analysis

The structural properties and phase purity of a-C:H:Cr coatings were explored by XRD using PANalytical X'Pert PRO diffractometer (Malvern Panalytical, Malvern, UK) in the 2$\theta$-$\omega$ configuration to remove the background caused by the silicon substrate. The angle between the incident X-ray beam and the detector forms an angle of 2$\theta$, and the $\omega$ angle represents the angle at which the X-ray beam hits the sample. Monochromatic Cu $-$ $k_\alpha$ Wavelength $\lambda = 0.15406$ nm has been used as the X-ray source. The device has been operated at 45 kV and 30 mA.

The microstructure of the a-C:H:Cr samples has been determined by TEM including selected area electron diffraction (SAED), high-resolution transmission electron microscopy (HRTEM), and energy dispersive X-ray spectroscopy (EDXS) on Thermo Fisher Tecnai (Thermo Scientific, Waltham, MA, USA) Osiris microscope operated at 200 kV.

The composition depth profile of the a-C:H:Cr coatings has been determined by X-ray photoelectron spectroscopy (XPS) depth profiling using a monochromatic X-ray source (Al K$\alpha$ radiation 1486.6 eV, spot size of 250 $\times$ 250 μm) using Ar$^+$ with an energy of 2 keV scanning over an area of 1.25 mm $\times$ 1.25 mm with incident angle of 30°, on ThermoFisher K-alpha spectrometer (Thermo Scientific, Waltham, MA, USA). The chamber pressure before the acquisition was below $9.9 \times 10^{-7}$ Pa. Each level of O 1s, Cr 2p, C 1s, and Si 2p (for substrate) is collected using snap mode (pass energy 147 eV, 10 snaps), with an erosion rate of 0.45 nm/s. The concentration is derived from each level, considering a Shirley background. A flood gun has been used during analysis, preventing eventual charging effects. No further energy shift is applied to the signal. The authors are aware of the recent warnings about XPS analysis [17,18]. Here, only the area of the peaks is considered, eventual energy shifts are not problematic. Approximately 30 at% of hydrogen was detected in each sample by the Elastic Recoil Detection technique (ERD) which is not considered in XPS data. All XPS data are computed using Avantage v.5.9916 Build 06625 software.

Raman spectroscopy was used to examine the structural variation of a-C:H:Cr coatings. The backscattering configuration using a LabRam HR 800 confocal laser system (Horiba, Kyoto, Japan) was selected with a laser wavelength $\lambda$ = 514 nm and a 2400 gr/mm grating. The spectral resolution for this configuration is ≈0.5 cm$^{-1}$. In order to avoid local heating and allow accurate measurement, the laser power was set to 10% of the maximum working power. All measurements were made without any polarization.

AFM measurements were performed in air, using the soft tapping mode of a Nanoscope III from Veeco Instruments (Santa Barbara, CA, USA). Images were recorded in 4 $\times$ 4 μm$^2$, and 2 $\times$ 2 μm$^2$ sizes with 512 $\times$ 512 lines per image and a scan rate of 1 Hz. The silicon

cantilever (Nanosensors PPP-NCHR, Nanosensors, Redwood City, CA, USA) with a resonance frequency of 290 kHz and a typical spring constant of around 42 N/m has been used for analysis. The tip has a nominal radius lower than 10 nm. The ratio between the set-point amplitude and the free amplitude of the cantilever vibration was always kept above 0.8. The average of three measurements for each sample has systematically been used. The topographic analysis of AFM images of the surface roughness was carried out using the Gwyddion software version 2.59.

The coating thickness, surface morphology, and cross-sectional growth were determined using a ZEISS Ultra 55 Field Emission Gun–Scanning Electron Microscope (FEG-SEM, Zeiss, Oberkoche, Germany). An electron high tension (EHT) of 5 kV in in-lens and secondary electron (SE) modes has been used to generate clear and less electrostatically distorted nano-structures images.

### 2.2.2. Nanoindentation and Micro-Scratch Tests

The hardness, Young's modulus, and apparent activation volume have been determined by nanoindentation using a G200 nanoindenter (KLA Tencor, KLA, Milpitas, CA, USA equipped with the Dynamic Contact Module (DCM) II head with 50 nN and 0.01 nm of force and vertical displacement resolution, respectively. A diamond Berkovich tip has been used and the tip area function has been calibrated using a fused silica reference. The nanoindentation measurements were performed at room temperature, under the load-control mode, with an exponential loading in order to produce a constant strain rate of $\dot{P}/P = 0.05 \, \text{s}^{-1}$. The thermal drift rate has been limited to 0.05 nm s$^{-1}$ before each experiment to ensure a negligible impact on the measured displacement. Sixteen indents were made within each sample for statistical analysis and consistency inspection. The measurements were carried out using the continuous stiffness measurement (CSM) technique providing continuous hardness and Young's modulus with increasing indentation depth. The hardness and modulus were calculated using the Oliver and Pharr model [19]. Also, Young's modulus of the coating was corrected from the substrate effect using the model proposed by Hay and Crawford [20]. In order to extract the apparent activation volume, constant contact stiffness relaxation tests have been performed with a holding duration of 30 min and a maximum penetration depth of 90 nm. According to Gu et al. [21], the activation volume $V^*$ is extracted from nano-indentation as follows:

$$V^* = 3\sqrt{3}kT\frac{\partial ln\dot{\varepsilon}_i}{\partial H},\tag{1}$$

where $\dot{\varepsilon}_i$ stands for the indentation strain rate, $k$ represents Boltzmann's constant, and $T$ is the testing temperature. The corresponding experimental and data processing procedures are explained in more detail in [22].

Micro-scratch experiments were performed using a G200 nanoindenter (KLA Tencor) with a XP head equipped with a lateral force measurement. This setup allows the measurement of the coefficient of friction during scratch. The XP head has a maximum normal force of 500 mN. A sphero-conical diamond tip with a radius of 5 µm and 90 degrees cone equivalent angle was used to perform the tests. Experiments involve applying a linear axial loading from 0 to 100 mN while sliding the indenter tip over 600 µm with a velocity of 30 µm·s$^{-1}$. The sample surface along the scratch path was scanned at low load (20 µN) with the indenter tip before and after the experiment to characterize the surface roughness and topology before, and measure the residual profile afterward. For each sample, a set of 8 scratches has been performed. Scratch grooves were observed by SEM to characterize the failure mechanisms along the wear track.

### 2.2.3. Tensile Test on Polymer Substrate

The fracture toughness of a-C:H:Cr coatings has been determined by deforming the coatings deposited on polymer substrates up to cracking. Prior to coating deposition, the polymer sheet was cut into tensile specimens using a punch. The geometry of the

specimens is shown in Figure 2. In-situ optical microscope and in-situ SEM tensile tests were performed with a Gatan micro testing machine with a maximum force of 2 kN and displacement rate of 0.1 mm/min. All experiments were performed at room temperature.

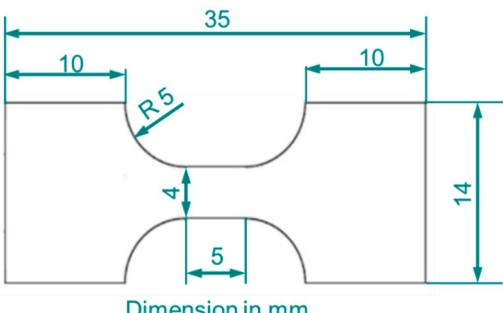

Dimension in mm

**Figure 2.** Specimen geometry for tensile testing of a-C:H:Cr coatings on kapton.

The cracking behavior was characterized in-situ by SEM, while in-situ optical microscope tensile tests were used to characterize the fracture toughness of the coatings. In-situ SEM tests require interrupting the test for imaging, which leads to some spurious creep in the polymer substrate. It has been shown in the literature that test interruption can lead to crack propagation even when the mode I stress intensity factor ($K_I$) remains below the critical value ($K_{Ic}$) of the coating [23]. Meanwhile, in-situ SEM tensile tests have been used to accurately determine the saturation crack density ($\rho_{sc}$), i.e., the maximum crack density reached when pulling on the specimen. In-situ SEM tests were initially stopped every 50 µm of crosshead displacement until 400 µm in order to take images at various magnifications. After the first 400 µm of displacement, the experiment was stopped every 100 µm of displacement until a total crosshead displacement of 1 mm. Two images per second were acquired during in-situ optical microscope tensile test at a magnification of 200× with a resolution of 1272 × 952 pixels which ensures looking at the whole specimen gauge length while seeing the cracks. These parameters are well suited to observe crack propagation and calculate the critical fracture energy release rate $G_{Ic}$ from

$$G_{Ic} = \frac{Z\sigma_f^2 h_f}{E'_f} ,\tag{2}$$

where $Z$ is a dimensionless parameter that depends on the elastic mismatch between the coating and substrate, $\sigma_f$ is the fracture stress in the coating, $h_f$ is the coating thickness and $E'_f = E_f/(1 - \nu^2)$ is the plane strain of Young's modulus. The $Z$ factor depends on the Dundurs parameter $\alpha$ as defined in [24] and on the substrate/coating thickness ratio [25]. The stress inside the coating is obtained from Young's modulus and from the strain at which the first crack is propagating, assuming the coating behaves as a linear elastic material and using Young's modulus extracted by nanoindentation. This will turn out to be a reasonable hypothesis with respect to the level of strain that is reached when looking at the literature [26,27]. The fracture strain ($\varepsilon_f$) is measured by following the displacement of small asperities on the images located close to the end of the gage region of the specimen. Note that the residual stress of a-C:H:Cr is considered equal to zero because the soft polymer substrate rapidly relaxes this stress, contrary to the condition impressed by a Si substrate. Finally, Equation (2) can be rewritten as

$$G_{Ic} = Z h_f E'_f \varepsilon_f^2,\tag{3}$$

and the related stress intensity factor $K_{Ic}$ as

$$K_{Ic} = \sqrt{G_{Ic} E'_f} ,\tag{4}$$

Despite the possibility of extremely high fracture toughness, the relatively high yield strength causes a reduction in the size of the fracture process zone, $d$, ahead of the crack tip [28]. The fracture process zone size, $d$, is defined as

$$d = \frac{K_{Ic}^{2}}{\pi \sigma_{y}^{2}},$$
(5)

where $\sigma_{y}$ is the yield strength of the coating.

### 3. Results

#### 3.1. Structure, Composition and Surface Morphology

The microstructure of the a-C:H:Cr coatings under different deposition conditions (0.66 Pa, 2.66 Pa sputtering pressure, floating or −100 V bias voltage) have been studied by XRD and TEM analysis. X-ray diffractograms of the coatings detected no crystallinity in the coatings showing a fully amorphous nature of the coatings. The high-resolution micrograph and SAED pattern from TEM also confirm the amorphous character of the deposited coatings (see Figure 3).

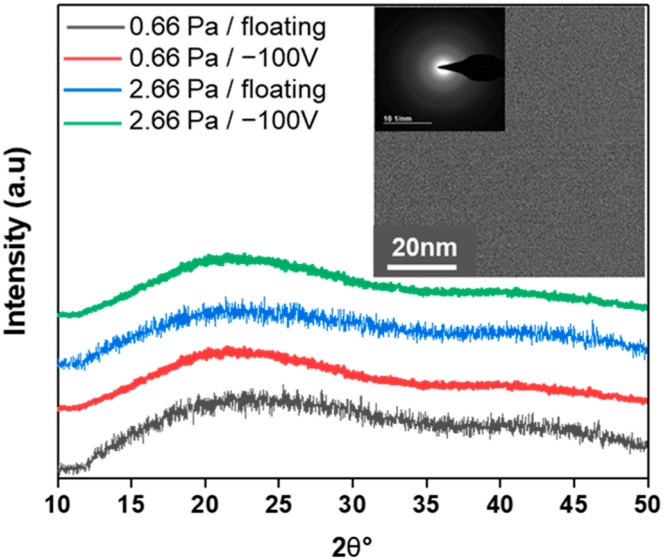

**Figure 3.** XRD pattern of the as-deposited amorphous coatings with different deposition conditions. The inset shows the high-resolution TEM micrograph and a diffraction pattern typical of amorphous coating.

Figure 4 displays the Raman spectra of the coatings. Each Raman spectrum was fitted into two Gaussian peaks. When applying a bias voltage, a small shift to the lower wave number is observed in the positions of the G peak ($\approx$12 cm$^{-1}$ and $\approx$6 cm$^{-1}$ for 0.66 Pa and 2.66 Pa, respectively). Also, an increase in FWHM of the G peak (from $\approx$112 to $\approx$130 cm$^{-1}$ and $\approx$110 to $\approx$122 cm$^{-1}$ for 0.66 Pa and 2.66 Pa, respectively), and D peak (from $\approx$334 to $\approx$339 cm$^{-1}$ and $\approx$328 to $\approx$334 cm$^{-1}$ for 0.66 Pa and 2.66 Pa, respectively), have been observed with the bias voltage application which is an indicator of more disordered amorphous coating. A decrease in the ratio of the D-band intensity to that of the G-band ($I_D/I_G$) and the G-band position are both indicators of lower hydrogen content and a rising sp$^3$ content, respectively [29,30].

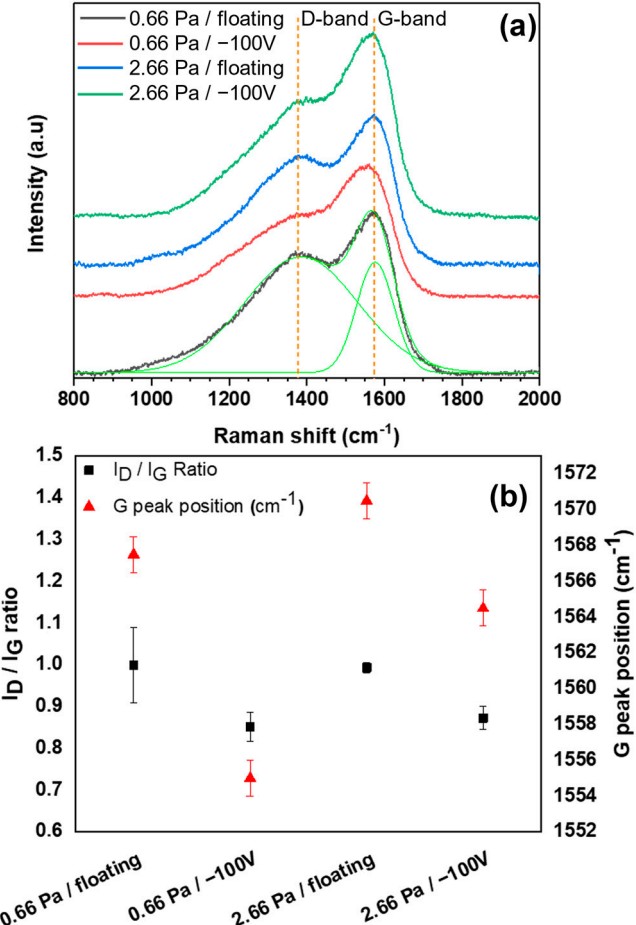

**Figure 4.** (**a**) Raman spectra of the a-C:H:Cr coatings. (**b**) Variation of the $I_D/I_G$ intensity ratio and the G peak positions under different deposition conditions.

The chemical composition of the coatings as a function of depth has been obtained by XPS analysis and ERD analysis. A comparison of the chemical composition of the coatings as a function of the applied bias voltage and deposition pressure is illustrated in Figure 5. The C content slightly increases, whereas the oxygen (O) content decreases with increasing applied bias voltage at a constant amount of Cr for both groups. The presence of higher oxygen content through the coating is associated with the low deposition power and bombarding of the substrate with lower energy than usual in order to prevent overheating of the Kapton substrate as well as the recrystallization of the a-C:H:Cr coatings. Furthermore, even though the base pressure in an industrial system like the one employed for this work is $1 \times 10^{-4}$ Pa, the oxygen concentration in the gas phase during deposition might be such that a small amount of oxygen is integrated into the films during the development due to the coater's walls outgassing [31]. The decrease in O content with increasing bias voltage is known as a preferential sputtering effect, which removes adsorbed oxygen during coating growth. When the bias voltage increases, the sputtering and etching energy of ions or atoms increases, and atoms' movement to the growing surface is enhanced. Due to the high deposition energy, C and Cr atoms are strongly bonded to the surface. Thus, weakly bonded O adatoms would be more easily re-sputtered by incident high-energy ions in the growth process of the coatings. This leads to dismissed O being replaced by C atoms [32] (extra information is listed in the Supplementary Materials).

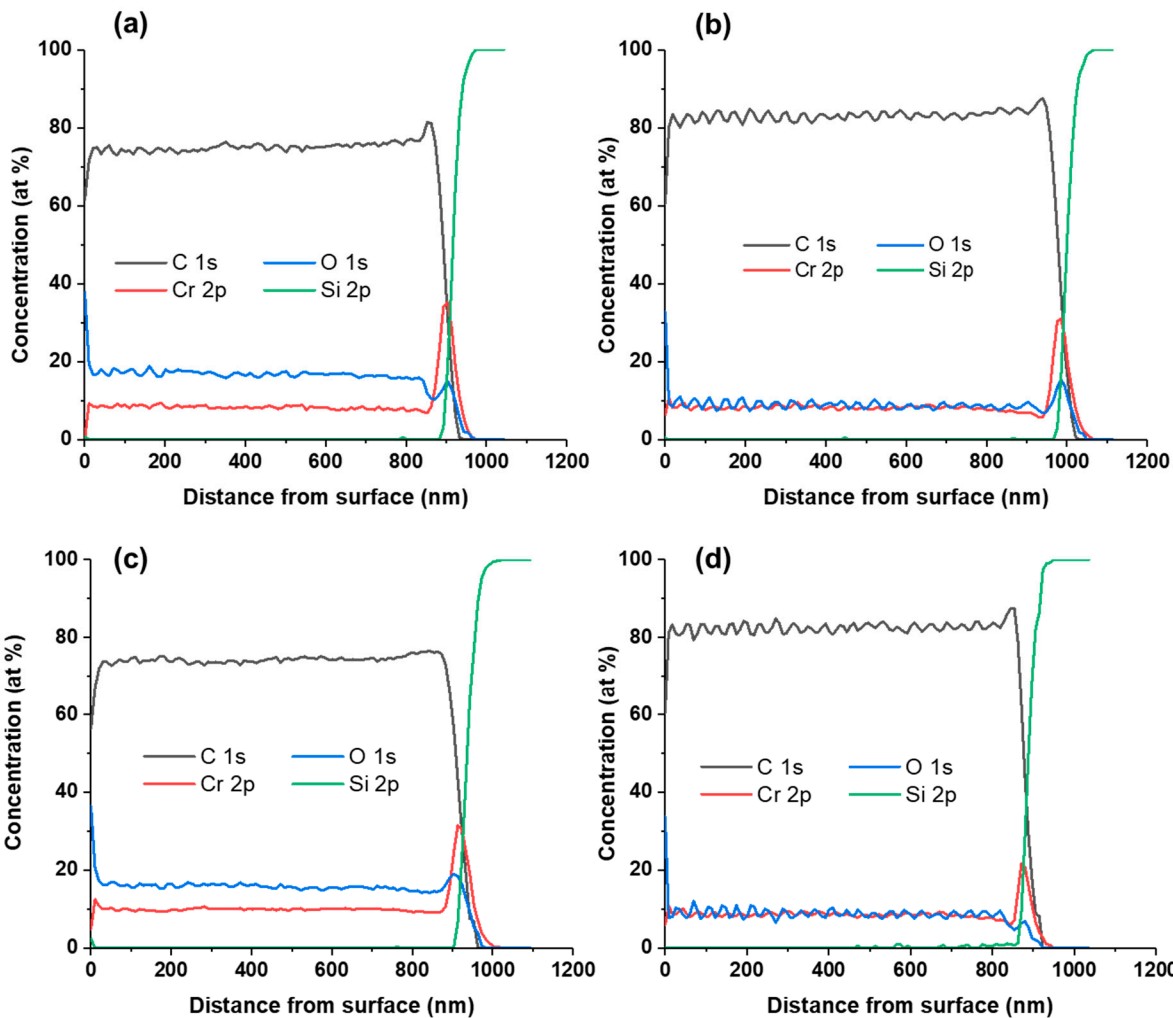

**Figure 5.** Chemical composition obtained by XPS depth profiling as a function of the coating thickness for (**a**) 0.66 Pa/floating and (**b**) 0.66 Pa/−100 V, (**c**) 2.66 Pa/floating, and (**d**) 2.66 Pa/−100 V conditions.

The AFM micrographs of the coatings over the $2 \times 2$ μm$^2$ area of selected regions are given in Figure 6. The roughness significantly depends on the applied bias voltage and working pressure [33]. A clear distinction can be made based on the bias voltage: the coatings obtained without bias exhibit a larger root mean square (RMS) roughness besides exhibiting columns wider than the one produced with −100 V bias. Also, the working pressure seems to affect the roughness when no bias is applied: increasing the working pressure leads to a smoother surface. Hence, the application of bias during deposition has a prominent effect on surface morphology. Table 1 presents the column diameter and surface roughness measured by AFM.

**Table 1.** a-C:H:Cr coatings mean column diameter and surface RMS roughness measured by AFM.

| Sample | Column Diameter (nm) | Surface RMS Roughness (nm) |
|---|---|---|
| 0.66 Pa/floating | 31.5 ± 2.6 | 7.8 ± 0.16 |
| 0.66 Pa/−100 V | 24.7 ± 0.7 | 6.2 ± 0.01 |
| 2.66 Pa/floating | 31.7 ± 1.6 | 6.7 ± 0.81 |
| 2.66 Pa/−100 V | 29.1 ± 1.6 | 6.2 ± 0.14 |

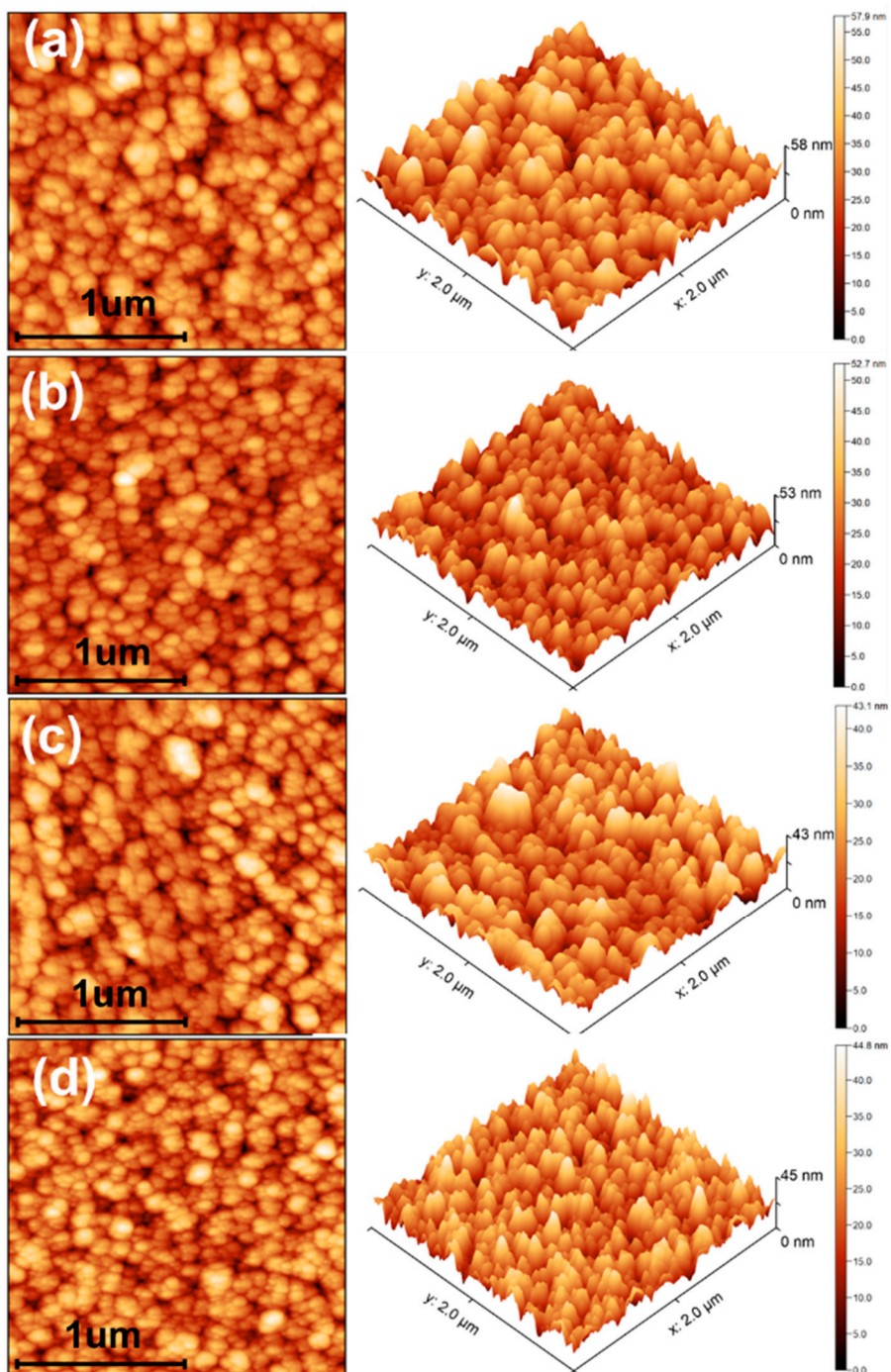

**Figure 6.** AFM images of a-C:H:Cr coatings (**a**) 0.66 Pa/floating and (**b**) 0.66 Pa/−100 V, (**c**) 2.66 Pa/floating, and (**d**) 2.66 Pa/−100 V.

The SEM top view and cross-section micrographs (Figure 7) reveal additional microstructure characterization, which varies with pressure and bias voltage. Cross-sectional views indicate the presence of a morphological texture with a columnar 'feature' (by analogy to nanocrystalline thin coatings) parallel to the growth direction. AFM top view observation and SEM cross-sectional view are in qualitative agreement in the case of columnar growth of a-C:H:Cr coatings.

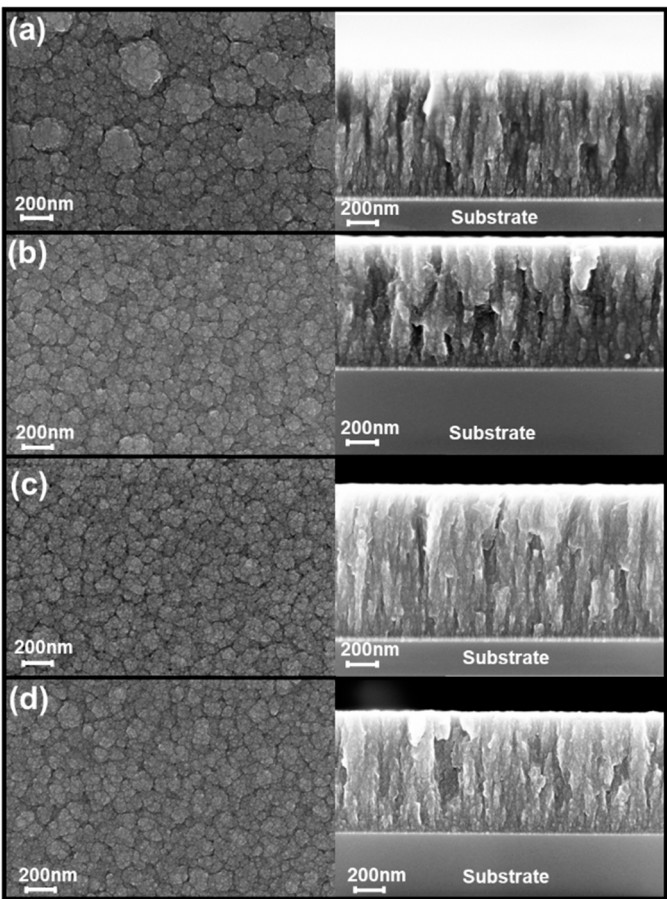

**Figure 7.** SEM micrographs on top views and cross-sections of a-C:H:Cr coatings: (**a**) 0.66 Pa/floating and (**b**) 0.66 Pa/−100 V, (**c**) 2.66 Pa/floating, and (**d**) 2.66 Pa/−100 V.

### 3.2. Mechanical and Tribological Properties

#### 3.2.1. Nano-Indentation

The hardness (*H*) and Young's modulus (*E'*) are provided in Table 2. Coatings deposited without applied bias voltage are softer with a hardness of 3.52 ± 0.44 GPa and 3.06 ± 0.37 GPa for 0.66 Pa/floating and 2.66 Pa/floating, respectively. With an increasing bias voltage, the hardness gradually increases to reach the highest value (4.95 ± 0.69 GPa) for the 2.66 Pa/−100 V sample. Since during deposition, the temperature did not exceed the room temperature (30 °C), no relaxation takes place in the coatings [34]. For instance, residual compressive stress on the order of 700–800 MPa has been determined using the Stoney method for the 0.66 Pa/−100 V sample deposited on the Si wafer [35]. The compressive stress experienced in many a-C:H:Cr coatings tends to close the cracks during indentation preventing crack initiation and propagation while simultaneously showing the smallest indentation imprint. All coatings illustrate the same trend in Young's modulus.

**Table 2.** Nano-indentation and micro-scratch related results: hardness, Young's modulus, resistance to plastic deformation, wear resistance, and critical load values for different deposition conditions.

| | $H$ (GPa) | $E'$ (GPa) | $\frac{\sigma_y}{E'} \times 10^{-2}$ | $\frac{\sigma_y^{\,3}}{E'^2} \times 10^{-3}$ (GPa) | $L_c$ (mN) |
|---|---|---|---|---|---|
| 0.66. Pa/floating | 3.5 ± 0.4 | 36.2 ± 2.8 | 5.21 ± 0.57 | 5.12 ± 0.56 | 50 ± 1.3 |
| 0.66 Pa/−100 V | 4.7 ± 0.8 | 45.1 ± 4.3 | 5.80 ± 0.81 | 8.80 ± 0.12 | 63.7 ± 2.0 |
| 2.66 Pa/floating | 3.1 ± 0.4 | 32.3 ± 1.9 | 5.15 ± 0.51 | 4.40 ± 0.43 | 48.4 ± 10.7 |
| 2.66 Pa/−100 V | 4.9 ± 0.7 | 48.7 ± 3.9 | 5.60 ± 0.65 | 8.37 ± 0.99 | 57.7 ± 1.2 |

The performance indices $\sigma_y/E'$ and $\sigma_y^3/E'^2$ extracted from nanoindentation analysis are shown in Figure 8 and Table 2 as indicating the resistance of wear behavior and plastic deformation of coatings. The yield strength is estimated from the hardness (*H*) as

$$\sigma_y = \frac{\xi_3 \tan(\beta) H}{\xi_1 \tan(\beta) - (1 - \xi_2)\frac{H}{E'}} \, , \tag{6}$$

where $\xi_1$, $\xi_2$, $\xi_3$, and $\beta$ values for a Berkovich tip are equal to 0.66, 0.216, 0.24, and 19.7, respectively [36].

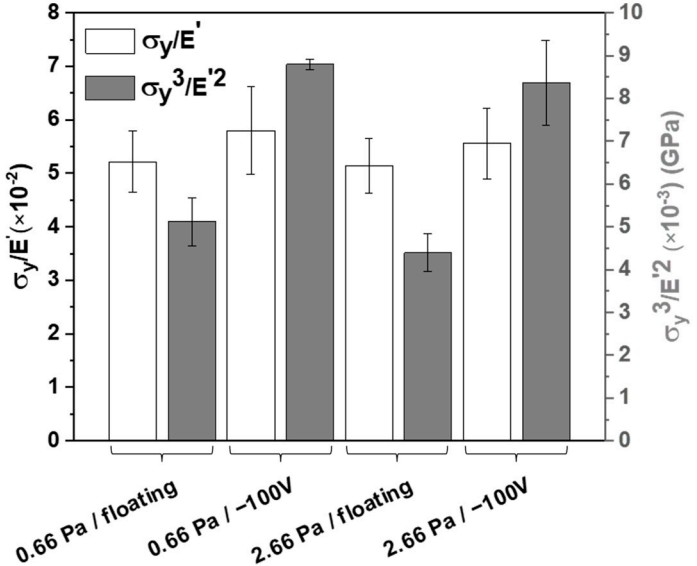

**Figure 8.** Comparison of the performance indices: $\sigma_y/E'$ and $\sigma_y^3/E'^2$ for the four conditions investigated in this study. The error bars represent the standard deviation calculated over 16 experiments.

The apparent activation volume $V^*$ can be related to the physical activation volume $\Omega$ as $V^* = \Omega\gamma$ where $\gamma$ is the transformation strain associated with the local atomistic shuffling mechanism [37], which is the deformation mechanism expected for this class of amorphous solid. The value of $\gamma$ for amorphous systems ranges from 0.05 to 0.15, see Argon [38]. We assume a transformation strain $\gamma = 0.1$ to obtain a rough estimate of the elementary volume in which the permanent elementary plastic deformation mechanism takes place. Figure 9 shows the variation of physical activation volume and the number of atomic sites involved in this volume as a function of the deposition parameters. The physical activation volumes obtained from relaxation tests also illustrate the same trend as for the nanoindentation results. Lower physical activation volume corresponds to coatings with larger hardness.

The lowest physical activation volume is found for the 2.66 Pa/−100 V thin coating as equal to $\Omega = 0.85 \pm 0.11$ nm$^3$. When no bias is applied, the working pressure does not significantly affect the physical activation volume which is around $1.5 \pm 0.23$ nm$^3$.

### 3.2.2. Wear Behavior

The wear behavior of the coatings has been directly evaluated by measuring the critical load at the onset of cracking ($L_c$) as shown in Figure 10a. $L_c$ is defined as the load at which the coating fails under scratching conditions. Elastic recovery is also displayed as a function of sliding distance in Figure 10b. Total elastic recovery corresponds to 1 and fully plastic deformation corresponds to 0. 0.66 Pa/−100 V condition remains nearly fully elastic until it fails. The variation of $L_c$ for each material is displayed in Figure 10c. The bias voltage plays a major role in the enhancement of the coating resistance to wear. Also, the deposition working pressure seems to be significant, with the 0.66 Pa/−100 V sample

being clearly more resistant to the onset of wear compared to the 2.66 Pa/−100 V one. The coefficient of friction is also an important index of the performance of an industrial coating (see Figure 10d). It must remain low even under high load.

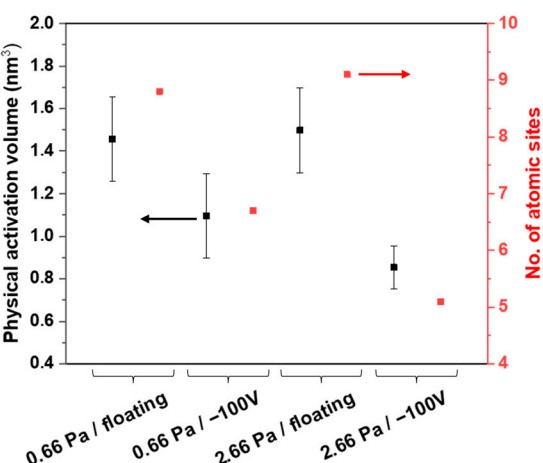

**Figure 9.** Effect of the deposition parameters on the physical activation volume and associated number of atomic sites of a-C:H:Cr thin coating. Arrows are for the identification of y-axes.

The critical load for the onset of chipping increases by applying a bias voltage, see Figure 10 and Table 2. Whatever the working pressure, applying a bias voltage leads to an increase in the wear resistance of the a-C:H:Cr coatings. Figure 11 displays typical wear tracks produced by micro-scratching. Several features distinguish the materials failure modes: (i) the first chip forms at a critical load noted $L_c$ propagates both forward and backward; (ii) the chip formation mechanism varies from random to reproducible chip sizes and shapes; (iii) a step-like pattern is evidenced along the crack surface. For 2.66 Pa coatings, the first chip propagates at a significantly lower critical load compared to 0.66 Pa coatings. This would imply that delamination at the interface between the Si wafer and a-C:H:Cr coating begins earlier—at a lower load—for the 2.66 Pa samples. The shape and the size of the chips depend on the bias voltage applied during deposition. Indeed, without bias, the chips are relatively diverse while for biased samples (−100 V), the chips are very reproducible. Also, for biased samples, chips are smaller and crack propagation is limited [10,39]. Finally, the fracture surface of the chip presents a step-like pattern as highlighted in Figure 11f. A qualitative counting of these features gives 20 occurrences. This is in line with the number of rotations performed during deposition (~23)—the samples being either close to the acetylene plasma or away. By getting far from the target, the risk of oxidation increases due to the absence of energetic ions bombarding the surface along the outermost surface of the coating, which produces a weaker interface for the next layers.

### 3.3. Uniaxial Tensile Tests

In-Situ SEM Tensile Tests

In order to determine the saturation crack density ($\rho_{sc}$), 200× magnification SEM micrographs have been selected—i.e., with a typical 580 µm linear field of view to ensure statistically relevant measurements. Figure 12a–d shows crack patterns at 1 mm of crosshead displacement (at 1000× magnification in order to distinguish the cracks clearly) when saturation is reached. Increasing the bias voltage and decreasing the pressure leads to lower crack density and earlier saturation of the cracking process, see Table 3. Figure 12e shows a typical crack at very high magnification in a 2.66 Pa/−100 V thin coating. The cracks systematically propagate along the cluster boundary.

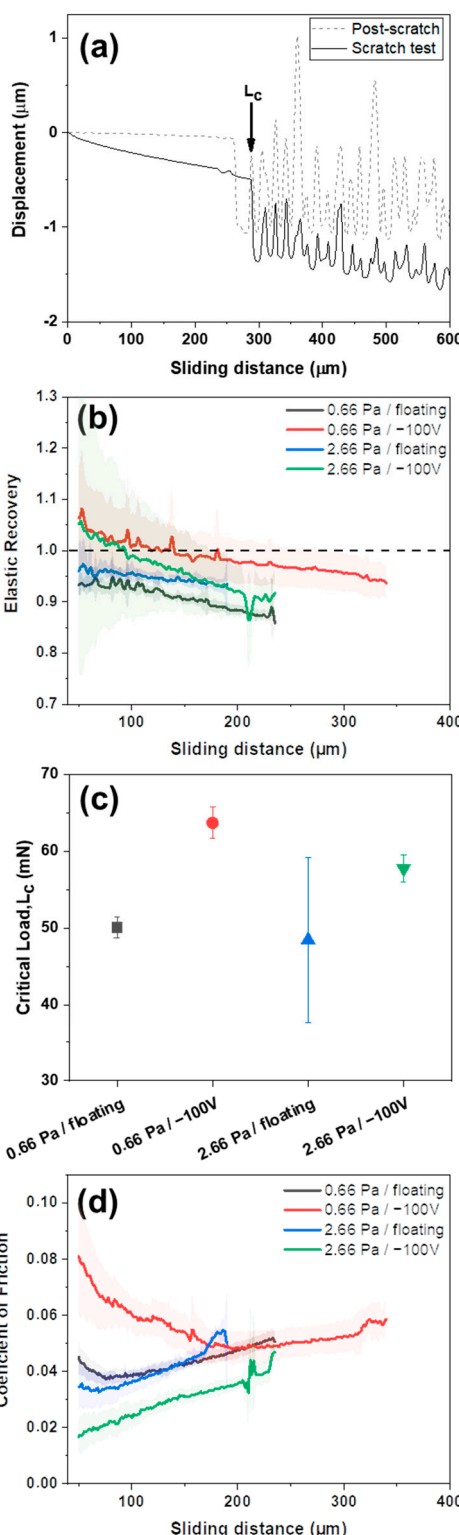

**Figure 10.** Results from the nanoscratch tests: (**a**) typical scratch profile with the displacement during loading as a black plain line and the profile of the wear track after loading in a dashed black line. The black arrow indicates the location of the critical load; (**b**) variation of the elastic recovery as a function of the sliding distance; (**c**) average critical load measured for each coating, error bars indicate the standard deviation; (**d**) variation of the coefficient of friction as a function of the sliding distance.

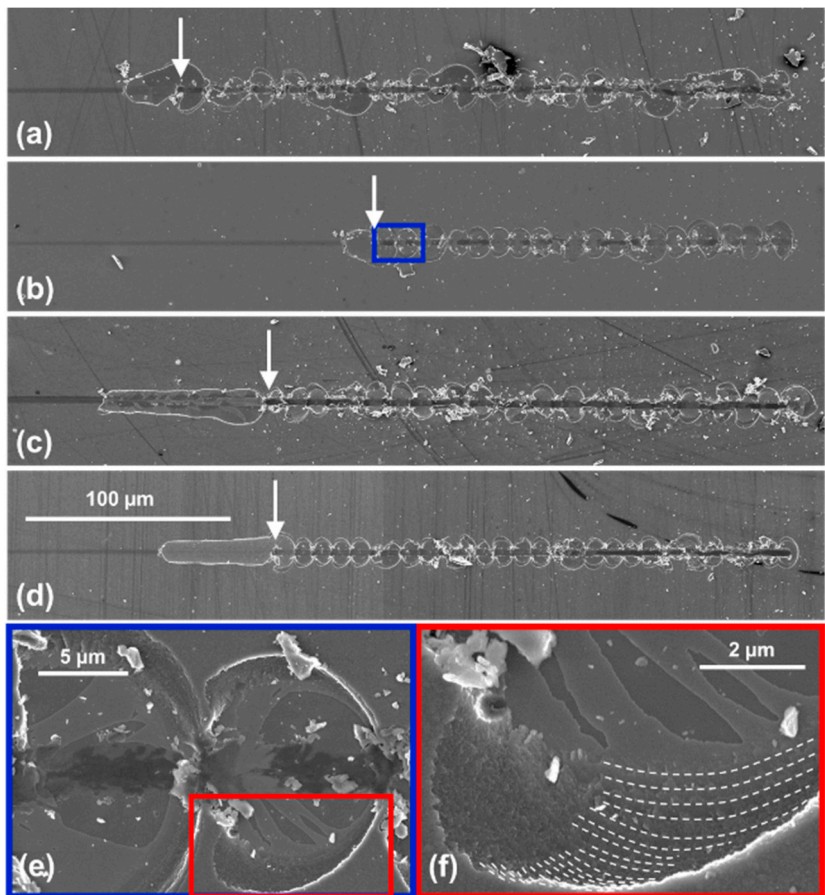

**Figure 11.** Typical wear track after nano-scratch testing of (**a**) 0.66 Pa/floating and (**b**) 0.66 Pa/−100 V, (**c**) 2.66 Pa/floating, and (**d**) 2.66 Pa/−100 V. The white arrows indicate the location of the critical load. The blue box highlights in (**e**) the formation of a chip just after the critical load on a 0.66 Pa/−100 V sample. The red box displays in (**f**) the crack features of the chip. The dashed white lines highlight the step-like pattern produced during the formation of the chip.

**Table 3.** Traction related results: Saturation crack density at 1 mm of crosshead displacement, critical fracture energy release rate, fracture toughness, and fracture plastic zone size values for different deposition conditions.

|  | $\rho_{sc}$ ($\mu$m$^{-1}$) | $G_{Ic}$ (J/m$^2$) | $K_{Ic}$ (MPa·m$^{1/2}$) | $d$ (nm) |
|---|---|---|---|---|
| 0.66 Pa/floating | 0.128 ± 0.004 | 4.94 ± 0.04 | 0.42 ± 0.08 | 15.7 ± 0.82 |
| 0.66 Pa/−100 V | 0.049 ± 0.001 | 47.4 ± 0.83 | 1.46 ± 0.05 | 100.8 ± 1.08 |
| 2.66 Pa/floating | 0.173 ± 0.002 | 125 ± 5.63 | 2.01 ± 0.18 | 488.6 ± 2.68 |
| 2.66 Pa/−100 V | 0.1 ± 0.003 | 115 ± 2.95 | 2.37 ± 0.12 | 242.5 ± 1.55 |

The fracture toughness of the coatings has been determined using Equation (4) with a Z factor equal to 15.6, as computed by [25] for a ratio $H/h$ around 30 and a $\alpha$-Dundurs parameter equal to 0.9. The plane strain Young's modulus is obtained from nanoindentation experiments. The strain has been determined from in-situ optical microscopy tensile tests by taking the strain at which the first cracks start to propagate. $K_{Ic}$ values of the different coating conditions are presented in Table 3 as well as the corresponding $G_{Ic}$. Increasing the working pressure leads to higher fracture toughness, as well as applying a bias voltage. Also, the application of a bias voltage decreases the crack density at saturation $G_{Ic}$ values above 100 J/m$^2$ which is an excellent resistance to cracking for a 1μm thick coating [40].

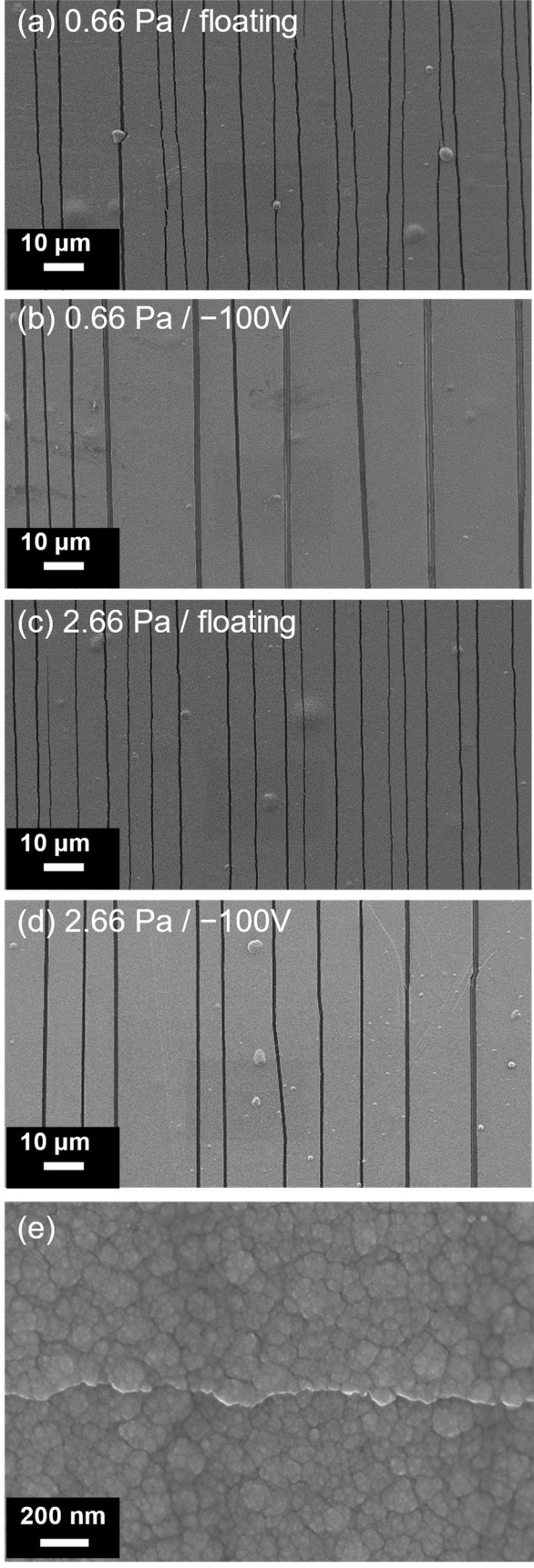

**Figure 12.** (**a–d**) Evolution of crack density with in-situ strain for different samples (**e**) typical high magnification SEM top-view observation of a crack.

## 4. Discussion

*4.1. Effect of Process Parameters on Hardness, Young's Modulus, and Activation Volume*

The magnitude of the hardness and Young's modulus are in good agreement with similar a-C:H coatings from the literature. For instance, Wei et al. [41] measured hardness between 3–6 GPa, Kassavetis et al. [42] between 2–6 GPa, Vanden Brande et al. [43] around 4 GPa, and De vriendt et al. [44] around 7 GPa. Also, in these studies, Young's modulus varies between 22.7–49.2 GPa, 20–45 GPa, 25–30 GPa, and 70 GPa, respectively.

By increasing the deposition pressure, ions are more likely to react with acetylene or to recombine with low-energy electrons. This leads to a reduction in the calculated ion mean free path from ≈22 mm to ≈5 mm and a rise in the number of collisions from $1.8 \times 10^4$ collision/s at 0.66 Pa up to $7.2 \times 10^4$ collision/s at 2.66 Pa. By knowing the target to substrate distance (≈150 mm) one can assume that number of collisions will increase from $1.2 \times 10^5$ to $2.1 \times 10^6$ collision/s from 0.66 Pa to 2.66 Pa, so it will decrease the energy of incident ions much more by increasing the deposition pressure. These reactions and recombination reduce the amount of energetic $C_2H_2^+$ and $Ar^+$ reaching the substrate. As a result of the sub-plantation process being reduced, energetic ion penetration into the growing coating will be prevented, which will lead to less $sp^3$ bonding, which is a crucial characteristic of hard coatings [15,16,45,46]. Consequently, coating growth results from the condensation of low energetic species. In this case, growth will mostly be caused by the condensation of $C_{2n}H_3$ molecules which leads to a decrease in the hardness value [15,16]. Amorphous carbon coatings deposited from $C_aH_b$ base without applied bias voltage are known to contain a higher density of C-H bonds which involve much smaller displacement energy than C-C bonds as present in a-C:H coatings [16], and coatings will contain a high amount of free volumes in their structure. When the bias voltage is applied, an increase in the energy of the ions impinging the substrate causes electron and carbon atoms to yield to compensate for free volume deficiency. This has been confirmed by the Raman spectroscopy results presented in Figure 4b and hardness values where the $I_D/I_G$ decreases with increasing $sp^3$ content at lower pressure and biased coatings. Although analyzing the $I_D/I_G$ for a-C:H:Cr does not precisely reflect the hardness variation since the specimens are made up of two different elements rather than only carbon, the results may typically be used to identify patterns based on carbon structure. The increase in the coating hardness is associated with the decline in $I_D/I_G$ [11].

There is also a relation between the applied bias voltage and the physical activation volume. Lower activation volumes are found when a bias voltage is applied during deposition, this would be another evidence of higher packing density [47]. Furthermore, assuming that the interatomic spacing of the surrounding C atoms is similar to the one of graphite, i.e., 0.34 nm [48,49] and that these sites present a spherical shape, we can estimate the volume of each site to be ≈0.16 nm$^3$. These characteristics relate to crystalline C and do not represent the free volume of the amorphous structure of a-C:H:Cr. Nonetheless, the physical activation volume obtained from the nanoindentation test corresponds to 5 to 9 crystalline C atom sites. This suggests that the elementary volume where plastic deformation events occur is very localized and includes the motion of a few C atoms only. The lower the physical activation volume, the less the atoms will be involved in the viscoplastic deformation mechanisms which is favorable in terms of the mechanical stability of the coatings owing to higher strain rate sensitivity. Since the physical activation volume has the lowest value for biased samples, a stronger atomic arrangement of carbon clusters is expected as can also be seen from the elastic properties. The present results confirm that the effect of the bias voltage is predominant over the effect of deposition pressure.

*4.2. Effect of Process Parameters on Wear Resistance*

The critical load $L_c$ at which scratch damages start appearing on the coating is used to qualify the wear resistance with a similar tip. In addition, the coating/substrate adhesion, the detachment of the fragments is also related to the cohesive (within the coating) fraction. The critical load has been shown to be proportional to the ratio $\sigma_y / E'$ quantifying the

ability to deform elastically [50], which is in good agreement with what is observed here when applying a bias voltage (see Table 2). Again, applying a bias voltage results in enhanced scratch resistance. This is a known result, but this scenario is also in good agreement with the activation volume values. Note that even though the activation volume for 2.66 Pa/−100 V is lower than 0.66 Pa/−100 V, the critical load of the coating deposited under 0.66 Pa/−100 V is a bit higher. This can be explained by the weak interface adhesion between the Si substrate and the 2.66 Pa sample as compared to the adhesion of the 0.66 Pa coating. The key information extracted here is that a lower activation volume and higher $\sigma_y/E'$ and $\sigma_y^3/E'^2$ leads to a longer elastic track on the coating, delays the plastic deformation, and increases the critical load to the onset of crack.

Table 4 compares our results with other studies using the same deposition technique in the literature. Results are in good agreement with coatings made of comparable a-C:H:Me materials found in the literature. The choice of the target is important. Additionally, our results based on a metallic target and $C_2H_2$ precursor gas show a higher critical load than one obtained for coatings deposited from sputtering a graphite target.

**Table 4.** a-C:H:Cr coatings deposited by PECVD method with $CH_4$ and $C_2H_2$ precursor gases or graphite target and nano-indentation results in comparison with recent work.

| Sample | Precursor Gas/Source | $H/E'$ | $L_c$ (mN) |
|---|---|---|---|
| Ref. [51] | $CH_4$ | 0.124–0.134 | 33–64 |
| Ref. [52] | $CH_4$ | 0.11–0.16 | 43–120 |
| Ref. [53] | $CH_4$ | - | 18.2–23.5 |
| Ref. [54] | $C_2H_2$ | 0.041–0.086 | 36–75 |
| Ref. [55] | $C_2H_2$ | 0.081 | 68 |
| Ref. [56] | $C_2H_2$ | 0.067 | 2.13 |
| Ref. [57] | Graphite | 0.14 | 6 |
| Ref. [58] | Graphite | ≈0.083 | 1.9–5 |
| Ref. [59] | Graphite | ≈0.081 | 40 |
| Our study | $C_2H_2$ | 0.09–0.1 | 48–64 |

*4.3. Effect of Process Parameters on Fracture Behavior*

The SEM micrograph of Figure 12e shows that the preferential crack path follows the boundary of columns. This highlights weaker inter-column boundaries. This is comparable to nano-glasses with weaker grain boundaries in comparison with the glassy matrix because of the higher free volume density contained in the boundaries [60]. However, the cluster interface strength also depends on the applied biased voltage. A decrease in columns' size is found in samples with bias voltage, thus for the smaller clusters—i.e., a higher ratio of the interface to cluster—the crack resistance is the highest. If the interface strength is constant, a lower number of interfaces will lead to an enhanced crack resistance, whatever the applied bias. Indeed, increasing the bias voltage leads to an increase in the adatom energy. Thus, atoms will attach more firmly to the substrate and already present clusters. It is therefore confirmed that the cluster interface is stronger when a bias is applied, providing enhanced crack resistance [61].

Finally, the $\sigma_y/E'$ a performance indicator is plotted against the fracture energy ($G_{Ic}$) to compare the four different a-C:H:Cr coatings to an extensive list of thin coating materials gathered from the literature [62–78] in Figure 13. As an indicator of cracking resistance, the a-C:H:Cr coatings present the largest yield strength over Young's modulus ratio with tunable $G_{Ic}$ in the range of 5–125 J/m$^2$.

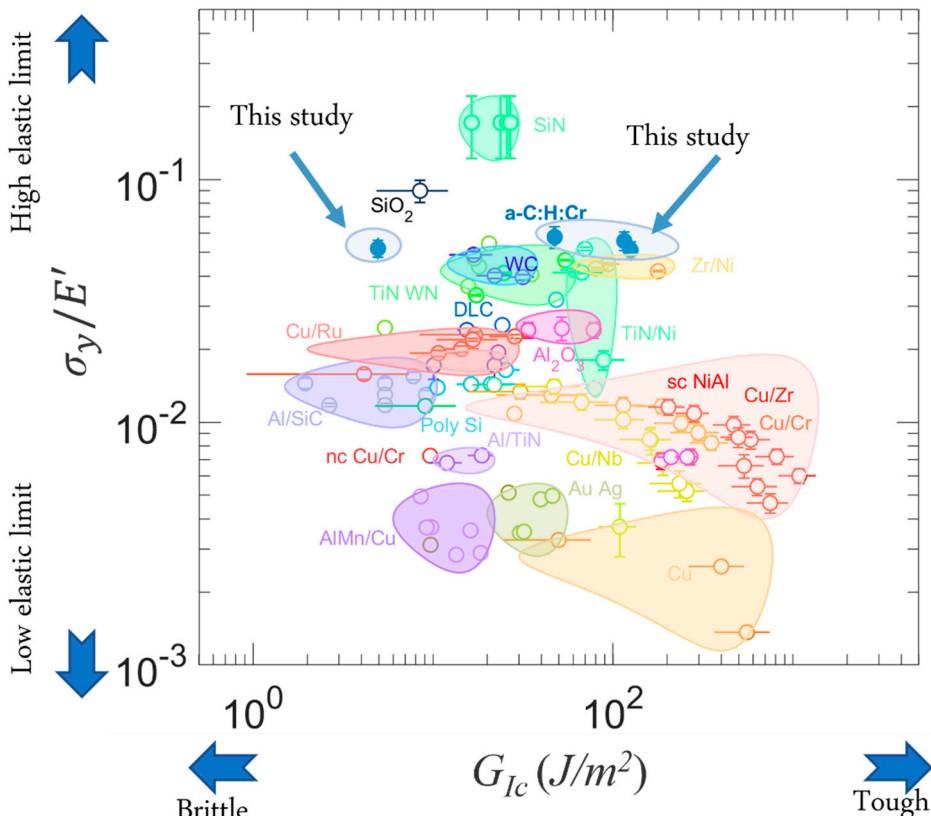

**Figure 13.** Materials property map presenting the ratio yield strength over Young's modulus against critical energy release rate. The results are obtained from the literature and compared to the four a-C:H:Cr thin coatings produced in this study [62–78].

Higher $G_{Ic}$ and $K_{Ic}$ values for 2.66 Pa samples can be directly related to their larger fracture strain and plastic process zone size which is more than two times larger than in 0.66 Pa coatings. Thus, by increasing the plastic process zone size, due to enhanced crack tip plasticity, the crack tip will be more blunted, and more energy will be needed to grow damage in the fracture process zone before material separation [79].

Accordingly, a-C:H:Cr coatings maintain a hardness that is substantially higher than that of elastomers while withstanding significant elastic deformation without showing signs of plasticity or failure. Also, the deposition conditions influence the coating toughness, and a relatively high critical energy release rate can be reached. This is of particular interest for applications where no residual scratches or imprints must remain after the contact. Protecting magnetic storage media from wear and corrosion due to their superior scratch resistance, bearings, gears, seals, engine components, oil media, and operation under ultra-high vacuum conditions are just a few of these applications [80–82]. Thus, according to the discussed properties, coatings with lower deposition pressure are attractive in applications where better adhesion and higher scratch resistance are needed. On the other hand, coatings with higher deposition pressure are beneficial for applications under extreme tension since they will exhibit larger fracture strain and fracture toughness.

## 5. Conclusions

The coating community often works to improve coating performances with different techniques using a trial-and-error approach. To the best of our knowledge, no rational material selection technique has been used yet to guide the optimization of a-C:H:Cr coatings based on targeted performance indices. In this work, we report a first effort in this direction to select the PECVD conditions in view of expected performances.

To this end, we evaluated the mechanical properties of four selected coatings deposited under different conditions (0.66 Pa, 2.66 Pa sputtering pressure, floating or $-100$ V bias voltage). Application of bias voltage was found dominant in terms of increasing the resistance to plastic deformation and wear resistance. Higher deposition pressure showed excellent resistance to cracking under tension, whatever the bias voltage.

A higher elastic limit (around 5%) was found when compared to a similar class of coatings in the industry while exhibiting tunable fracture toughness categorized in three different regimes of low ($<10$ J/m$^2$), mid (10–100 J/m$^2$), and high ($>100$ J/m$^2$) levels.

Finally, this study proves that instead of focusing on specific parameters, it will be more beneficial to mix and match the mechanical properties in order to meet different lists of requirements associated with different types of applications.

**Supplementary Materials:** The following supporting information can be downloaded at: https://www.mdpi.com/article/10.3390/coatings13122084/s1, Figure S1. XPS depth profiling maps illustrate for a-C:H:Cr coatings show the presence of each element versus the etching time.

**Author Contributions:** Conceptualization, E.H., T.P. and S.L.; Methodology, A.B., P.B., M.-S.C. and H.I.; Formal analysis, A.B.; Data curation, A.B., P.B., M.-S.C., A.O. and H.I.; Writing—original draft, A.B.; Writing—review & editing, A.B., P.B., M.-S.C., A.O., H.I., E.H., T.P. and S.L.; Supervision, E.H., T.P. and S.L.; Project administration, T.P. and S.L.; Funding acquisition, T.P. and S.L. All authors have read and agreed to the published version of the manuscript.

**Funding:** This research was funded by the Walloon region under the PDR FNRS project number C 62/5—PDR/OL, semaphore 33677636. H. Idrissi acknowledge the financial support by the Belgian National Fund for Scientific Research (FSRFNRS) under Grant CDR—J.0113.20. M.-S. Colla acknowledges the financial support of National Fund for Scientific Reaserch (FNRS), Belgium.

**Institutional Review Board Statement:** Not applicable.

**Informed Consent Statement:** Not applicable.

**Data Availability Statement:** Data are contained within the article and Supplementary Materials.

**Acknowledgments:** The Synthesis, Irradiation & Analysis of Materials (SIAM), Morphology and Imaging (Morph-IM), and Physico-Chemical characterization (PC2) platform of Unamur, LACaMi and WELCOME platforms of UCLouvain, and EMAT platform of UAntwerpen are acknowledged for IBA, AFM & XPS, XRD, SEM, Raman spectroscopy and TEM facilities.

**Conflicts of Interest:** The authors declare that they have no known competing financial interest or personal relationships that could have appeared to influence the work reported in this paper.

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
