# Peer review of "Tailoring Mechanical Properties of a-C:H:Cr Coatings"

_coatings, doi:10.3390/coatings13122084_

Round 1
Reviewer 1 Report
Comments and Suggestions for Authors
The technology of deposition of carbon films doped with metals, including carbide-forming ones, has been developing for more than thirty years. Various names were used for this kind of films - DLC-Me, a-C:H:Me, metal-dielectric diamond-like composites of atomic scale (DLASC), diamond-like nanocomposites (DLN), etc.
1. This work is devoted to the study of the structure and some mechanical properties of a-C:H:Cr coatings. Although the properties of the films obtained are far from record values for this class of materials, the integrated approach and detailed analysis of the results obtained are of scientific interest. The results obtained in the reviewed work complement the data of other authors published in several dozen papers. The main trend in the manufacture of a-C:H:Cr coatings is the formation of multilayer coatings and doping of a-C:H:Cr coatings with nitrogen from the gas phase during the deposition process. The proposal that “no rational material selection technique has been used yet to guide the optimization of a-C:H:Cr films” seems controversial.
1. The main disadvantage of this manuscript is that in the list of references contains only one article [15] from ninety (!) which is devoted to the deposition and study of a-C:H:Cr coatings. When finalizing the text of the manuscript, it is necessary to compare the results obtained with data from other authors, which also used magnetron sputtering for deposition of a-C:H:Cr coatings on different substrates. It is also advisable to reduce it a little the number of references which discuss the properties of coatings with significantly different chemical compositions. For example, (for your choice) - Weicheng, Kong, et al. "TEM structure, nanomechanical property, and adhesive force of magnetron-sputtered Cr-DLC coating on a nitrided Ti6Al4V alloy." The Journal of Physical Chemistry C 125.30 (2021): 16733-16745; Zou, Changwei, et al. "Further improvement of mechanical and tribological properties of Cr-doped diamond-like carbon nanocomposite coatings by N codoping." Japanese Journal of Applied Physics 55.11 (2016): 115501; Viswanathan, S., et al. "Corrosion and wear behaviors of Cr-doped diamond-like carbon coatings." Journal of materials engineering and performance 26 (2017): 3633-3647; Amanov, Auezhan, et al. "Fretting wear and fracture behaviors of Cr-doped and non-doped DLC films deposited on Ti–6al–4V alloy by unbalanced magnetron sputtering." Tribology International 62 (2013): 49-57; Zou, C. W., et al. "Effects of Cr concentrations on the microstructure, hardness, and temperature-dependent tribological properties of Cr-DLC coatings." Applied Surface Science 286 (2013): 137-141; Cui, Xue-Jun, et al. "Structure and anticorrosion, friction, and wear characteristics of Pure Diamond-Like Carbon (DLC), Cr-DLC, and Cr-H-DLC films on AZ91D Mg alloy." Journal of Materials Engineering and Performance 28 (2019): 1213-1225; Dai, Wei, et al. "Microstructure and property evolution of Cr-DLC films with different Cr content deposited by a hybrid beam technique." Vacuum 85.8 (2011): 792-797; Pal, Sunil K., et al. "Effects of N-doping on the microstructure, mechanical and tribological behavior of Cr-DLC films." Surface and Coatings Technology 201.18 (2007): 7917-7923, etc.
2. Statement that “Adding metal elements to the films can decrease internal stress and improve wear resistance, adhesion, thermal stability, and fracture toughness." should be supported by references to your own publications and/or papers of other authors.
3. It is necessary to supplement the manuscript with XPS spectra and provide an explanation for the presence of up to 20 at % oxygen in the bulk of the film. The phrase “The a-C:H:Cr films with 10 at% of Cr and around 30 at% of hydrogen” is in conflict with the data in Figure 5.
5. When analyzing Raman spectra, it is necessary to supplement the text with information about the FWHM values for D- and G-bands.
Reviewer 2 Report
Comments and Suggestions for Authors
see attached file

Comments on the Quality of English LanguageAs I wrote in the referee report, I only recommend the replacement of some terms
Reviewer 3 Report
Comments and Suggestions for Authors
In this article are presented studies about how the deposition parameters affect the mechanical characteristics (both effective and intrinsic) of thin a-C:H:Cr films deposited by magnetron sputtering. An influence of the bias voltage and Ar/C2H2 pressure were investigated. The optimization of strategy due to future application was conducted.
Presented work is on high scientific level and is well organised. Many measurement methods were used including X-ray Photoelectron Spectroscopy (XPS) to extract the coating’s chemical composition; Atomic Force Microscopy (AFM) and Scanning Electron Microscopy (SEM) to observe the structural and morphological properties of the films; Transmission Electron Microscopy (TEM) and X-ray diffractometer (XRD) to investigate the amorphous microstructure of the coating; Raman Spectroscopy to track structural variation; nanoindentation to determine the hardness, Young’s modulus, and activation volume; micro-scratch test to extract wear resistance, critical load, as well as tensile tests on a supporting polymer membrane to determine the fracture toughness of the coating.
The presented results are complex and well discussed.
Some minor improvement are needed:
1. Chapter numbering is wrong.
2. What was the distance between substrate and target mounted in magnetron? How was placed surface of substrate to target surface? It was perpendicular (rotating table fig.1) or vertical (rotating sample fig.1)? Has the angle of deposition influenced on properties of deposited films? What was the range of the loads using during nanoindentation tests?
3. Please describe in details the XRD measurement procedure. On the SEM images we see columnar structure with well-defined shape and columns thickness. It may suggest that such structure should be crystalline. Can the authors explain or discuss why SAED and XRD show amorphous structure?
4. How accurate is AFM measurement? In my opinion, providing measurement errors of roughness with an accuracy of two decimal places (nm) is an exaggeration.
5. The units of pressure should be in SI system - Pa against mTorr
6. In the figure 10b, d the shadow under curves is seen. Can you explain what this is?
7. Please check text again due to correction of editorial errors. Examples : line 411 – symbol “ ~ “before numbers; x axis description fig. 13 (J.m-2); unit of bias voltage and pressure are sometimes after space sometimes without space; and others…
